# Advocate-BREAST80+: A Comprehensive Patient and Advocate-Led Study to Enhance Breast Cancer Care Delivery and Patient-Centered Research in Women Aged ≥80 Years

**DOI:** 10.3390/cancers16142494

**Published:** 2024-07-09

**Authors:** Ciara C. O’Sullivan, Robert A. Vierkant, Nicole L. Larson, Mary Lou Smith, Cynthia Chauhan, Fergus J. Couch, Janet E. Olson, Stacy D’Andre, Aminah Jatoi, Kathryn J. Ruddy

**Affiliations:** 1Department of Oncology, Mayo Clinic, Rochester, MN 55905, USA; dandre.stacy@mayo.edu (S.D.); jatoi.aminah@mayo.edu (A.J.); ruddy.kathryn@mayo.edu (K.J.R.); 2Department of Quantitative Health Sciences, Mayo Clinic College of Medicine and Science, Mayo Clinic, Rochester, MN 55905, USA; vierkant.robert@mayo.edu (R.A.V.); larson.nicole2@mayo.edu (N.L.L.); olsonj@mayo.edu (J.E.O.); 3Research Advocacy Network, Plano, TX 75093, USA; mlsmith@researchadvocacy.org; 4Patient Advocate, Mayo Clinic Breast Cancer Specialized Program of Research Excellence (SPORE), Mayo Clinic, Rochester, MN 55905, USA; cynthiachauhan@aol.com; 5Department of Experimental Pathology & Laboratory Medicine, Mayo Clinic, Rochester, MN 55905, USA; couch.fergus@mayo.edu

**Keywords:** advocacy, breast cancer, older patients

## Abstract

**Simple Summary:**

The high-level aims of the Advocate-BREAST initiative are to study and improve the overall experience of patients with breast cancer (BC) through education, shared decision making, and patient-centered clinical trials. Advocate-BREAST80+ is a survey substudy that specifically focused on the unique needs and perspectives of BC patients aged ≥80 years. Although patients aged ≥80 years experienced less anxiety and symptom-related distress compared with younger patients, they were significantly less satisfied with information regarding short and long term side effects of BC therapies, as well as the management of same. Older patients were significantly less likely to have participated in a clinical trial or be open to considering this option in future. Future research should address unique educational needs and barriers to research participation in older BC patients. Focused interviews could assist with better comprehension of the lived experience of these patients, given the smaller number of BC patients ≥80 years in many available databases.

**Abstract:**

Background: There are limited evidence-based data to guide treatment recommendations for breast cancer (BC) patients ≥80 years (P80+). Identifying and addressing unmet needs are critical. Aims: Advocate-BREAST80+ compared the needs of P80+ vs. patients < 80 years (P80−). Methods: In 12/2021, a REDCap survey was electronically circulated to 6918 persons enrolled in the Mayo Clinic Breast Disease Registry. The survey asked about concerns and satisfaction with multiple aspects of BC care. Results: Overall, 2437 participants responded (35% response rate); 202 (8.3%) were P80+. P80+ were less likely to undergo local regional and systemic therapies vs. P80− (*p* < 0.01). Notably, P80+ were significantly less satisfied with information about the short and long-term side effects of BC therapies and managing toxicities. P80+ were also less likely to have participated in a clinical trial (*p* < 0.001) or to want to do so in the future (*p* = 0.0001). Conclusions: Although P80+ experienced less anxiety and symptom-related distress compared with P80−, they were significantly less satisfied with information regarding the side effects of BC therapies and their management. P80+ were significantly less likely to have participated in a clinical trial or be open to considering this option. Future studies should address educational needs pertaining to side effects and barriers to research participation in P80+.

## 1. Introduction

In 2023, approximately 297,790 American women were diagnosed with invasive breast cancer (BC) [1], and ~12% were aged ≥80 years [2]. Despite important advances in BC care, there are limited evidence-based data to guide treatment recommendations in patients ≥80 years (P80+), which can lead to both under and over-treatment [3], as well as limited data on these patients’ perspectives on cancer and cancer therapy. Unique challenges in this patient population include concerns regarding treatment-related toxicities, comorbidities and limited life expectancy, and underrepresentation in clinical trials [4]. Furthermore, although BC biology in older women is often indolent [5], older patients may experience worse BC-specific outcomes when controlling for disease subtype and stage [4,6]. Older women with BC may also differ from younger women in their educational needs, but there are limited data regarding these needs [7]. Online patient resources devoted to this population are limited [8]. Therefore, identifying and addressing gaps will be a critical step toward the optimization of care for older BC patients and should be a clinical and research priority [9].

The Advocate-BREAST project, “Advocates and Patients’ Advice to Enhance Breast Cancer Care Delivery, Patient Experience and Patient Centered Research”, is a collaboration between breast oncologists at Mayo Clinic in Rochester (MCR), Minnesota, USA and Mayo Clinic BC Specialized Program of Research Excellence (SPORE) advocates. Its high level aims are to study and improve the overall patient experience through education, shared decision-making, and patient-centered clinical trials. The primary objective is to identify areas of unmet need in BC care delivery and research with the goal of improving patient experience and driving further research. We initially conducted a patient experience survey in BC survivors enrolled in the Mayo Clinic Breast Disease Registry (MCBDR) [10]. This paper presents the results of Advocate-BREAST80+, a survey substudy that compares the perspectives of P80+ with BC patients < 80 years (P80−) after having experienced a diagnosis of BC.

## 2. Methods

### 2.1. Advocate-BREAST Project Approach and Committee Formation

The Advocate-BREAST project uses a five-step framework: Connecting, Comprehending, Analyzing, Implementing, and Reflecting (Figure 1). The overall goals are outlined in Figure 2. The Mayo Clinic project management team included two Breast Medical Oncologists [C.O.S. and K.J.R.], a Research Program Co-Ordinator [N.L.], a statistician [R.V.], and two senior Mayo Clinic SPORE BC advocates [M.L.S. and C.C.].

### 2.2. Study Design and Recruitment

The Mayo Clinic Comprehensive Cancer Center (MCCC) is an NCI-designated comprehensive cancer center with three main campuses in Minnesota (MCR), Florida and Arizona. We conducted a retrospective cohort study of patients enrolled in the MCBDR, a prospective longitudinal cohort study that enrolls patients with stage 0–4 BC who were seen at least once at MCR within one year of diagnosis (2001–2021). Informed consent was obtained prior to registry enrollment. Patients were excluded if they had received an initial BC diagnosis more than one year prior, did not speak English, or did not live within the USA.

### 2.3. Data Collection and Processing

In MCBDR, demographic information, histopathological tumor characteristics and treatment-related factors are abstracted from the electronic medical record by trained nursing personnel. For this study, we accessed age, sex, date of BC diagnosis and disease stage. Patients were classified into three groups: ductal carcinoma in situ (DCIS; stage 0), invasive non-metastatic BC (stage 1–3), and metastatic breast cancer (MBC; stage 4) at time of survey completion. ZIP codes of residence were mapped to rural–urban commuting area (RUCA) codes, and rurality was defined as RUCA code 10 (i.e., >60 min road travel (one-way) to the closest edge of an urbanized area) [11]. The Mayo Clinic Institutional Review Board (IRB1815-04) reviewed and approved this study. Data were handled in a manner consistent with both US laws and the Declaration of Helsinki.

In addition to medical chart review, participants completed a survey developed by the study team (see below) addressing satisfaction with (i) multiple aspects of cancer care delivery and (ii) the education and/or support they receive(d) regarding practical, financial, emotional, societal and spiritual concerns linked to their diagnosis. Racial/ethnic, educational, rural/urban, and financial status data were used to inform the development of novel resources to address patient-reported gaps in care. Participants were also asked to rank potential QI projects in order of the likelihood the proposal could improve quality of life for patients and their families. Patients were also asked to comment on how care for BC patients might be improved, and their thoughts on what research topics should be prioritized. Feedback from P80+ was initially reviewed by the research program coordinator, who categorized the content. The Breast Medical Oncology team further reviewed the qualitative data and classified each theme as either major or minor based on the number of responses. Comments regarding suggestions for further research were noted separately. Responses were collected via anonymous local language questionnaires.

### 2.4. Survey

Our 23-page survey, containing 147 items, was developed to assess patient levels of concern and satisfaction with various aspects of their BC diagnosis and treatment. We used REDCap, a secure web application for developing and managing online surveys and databases that is specifically tailored to support online and offline data capture for research studies. A questionnaire was electronically sent to all MCBDR participants who were consented, alive, and had an e-mail address on file on 12/9/2021. Non-respondents were sent two reminder e-mails on 12/16 and 12/23/2021.

The questionnaire captured: 1. Demographic Information; 2. BC Treatment; 3. Concerns Regarding Side Effects of BC Treatment; 4. BC Clinical Care Concerns ([i] level of symptoms experienced during the first year after BC diagnosis and [ii] level of concern regarding health-related, practical, financial, emotional, societal and spiritual issues related to BC during that time); 5. Clinical Care BC Patient Experience ([i] overall satisfaction with BC care, [ii] satisfaction with information and support received from the care team as regards symptom management during first year after BC diagnosis, and [iii] satisfaction with information and support received from BC care team as regards practical, financial, emotional, societal and spiritual issues related to BC during the first year after diagnosis); 6. Ranking of Proposed QI Projects; 7. Integrative Medicine; 8. Medical Second Opinions; 9. Clinical Trial Participation; and 10. Thoughts/suggestions (patient comments) on [i] how care for BC patients could be improved and [ii] what research topics should be prioritized in BC.

Sections 3–5 of the questionnaire contained different numbers of items to be scored, each on a 10-point Likert scale. For Section 3 and some items in Section 4, respondents were asked to score their responses (0 = not at all concerned, 10 = highly concerned). Other items in Section 4 related to symptom severity, and were scored accordingly (0 = none, 10 = [symptom] as bad as I can imagine). For Section 5, respondents were asked to rate their overall satisfaction with their BC care (0 = very dissatisfied, 10 = extremely satisfied). For the satisfaction data, scores of 0–3 were rated as low satisfaction, scores of 4–6 were rated as moderate satisfaction, and scores of 7–10 were rated as high satisfaction. Section 6 of the questionnaire asked respondents to rate the likelihood that a proposed QI project would improve patient care (0 = none, vs. 10 = as much as I can imagine).

### 2.5. Statistical Analysis

Participants were dichotomized into two groups based on age at survey completion: those younger than 80 (hereafter referred to as P80−) and those 80 or older (P80+). Data were summarized using frequencies and percents for categorical variables and means, standard deviations (SDs), medians and interquartile ranges (IQRs) for continuous variables. We compared demographic and clinical characteristics across survey response status using chi-square tests of significance. Amongst those who returned surveys, associations of survey responses with dichotomized age were first univariately examined using two-sample *t*-tests for continuous variables and chi-square tests for categorical variables. All statistical tests were two-sided, and all analyses were carried out using the SAS System (SAS Institute, Inc., Cary, NC, USA).

## 3. Results

A total of 6918 MCBDR participants (6877 female) were sent surveys, and 2450 responses (response rate = 35.4%) were received. Thirteen males were excluded, resulting in a final study size of 2437. Overall, the mean age of participants was 64 (SD 11.8), and 202 (8.3%) were aged ≥80 years. The mean time from BC diagnosis at the time of the survey was 93.3 months (SD 70.2). Comparisons of the demographic and clinical characteristics of P80+ vs. P80− across survey response status are provided in Table 1. Survey responses are listed in Table 2.

### Advocate-BREAST80+ Substudy Results

Demographic Information

Compared with responding P80−, responding P80+ were more likely to be white (*p* = 0.0136), widowed (<0.0001), living in a rural location (0.0136), and initially diagnosed with BC ≥ 24 months prior to completing the survey (<0.0001). For P80+, the mean time from BC diagnosis at the time of survey completion was 128.5 months (SD 74.4) compared with 90.1 months (SD 68.9) for P80− (*p* ≤ 0.001). The BC stage distribution for patients (Stage 0–4) was similar in P80+ compared to P80− (*p* = 0.2571). Based on these results, the following variables were used to calculate propensity scores for subsequent weighted analyses: race, marital status, religion, time since BC diagnosis, primary BC treatment location, and recommendation status for surgery, chemotherapy, radiotherapy, endocrine therapy, biologic therapy, and immunotherapy.

Treatment Recommendations

Overall, P80+ were less likely to report that they had been advised to undergo local regional and systemic therapies than P80−. Specifically, surgery was recommended less frequently for P80+ compared with P80− (*p* = 0.0062, Table 1). However, when surgery was recommended, P80+ were as likely to proceed with the same as P80− (*p* = 0.5643). Adjuvant radiation therapy was also less likely to be recommended for P80+ compared with P80− (*p* = 0.0037); however, there was no difference between the groups regarding the likelihood of proceeding with radiation therapy if advised (*p* = 0.6635).

Regarding systemic therapies, P80+ were less likely to be advised to pursue chemotherapy, immunotherapy, targeted biologic therapy or endocrine therapy compared with P80− (*p* ≤ 0.0001 for each of these treatments). Furthermore, although P80+ were less likely to proceed with recommended endocrine therapy (*p* = 0.0011), targeted biologic therapy (*p* = 0.0940), and immunotherapy (*p* = 0.0015), they were as likely as P80− to proceed with chemotherapy if recommended (*p* = 0.1641).

Symptom Severity

Overall, the most severe symptom experienced by patients in their first year following BC diagnosis was hair loss/thinning, followed by hot flashes, eyebrow/eyelash thinning, sexual dysfunction, cognitive/memory issues, neuropathy and lymphedema. Compared with P80−, P80+ were significantly less impacted by hot flashes, eyebrow/eyelash thinning, sexual dysfunction, cognitive/memory issues and lymphedema (*p* < 0.0001 each).

Level of Concern Regarding Issues within the First Year after Breast Cancer Diagnosis

The main concerns patients of all ages recalled in the first year following their BC diagnosis were (i) fear of BC recurrence/spread, (ii) concerns about loved ones coping, (iii) diagnosis and prognosis, (iv) fear of dying of BC, (v) weight/physical fitness, and vi) their emotional health. P80+ were significantly less impacted than P80− regarding these issues (*p* < 0.0001 for each item), although they were still identified as top concerns overall.

Overall Satisfaction with BC Patient Experience

Patients reported a high level of overall satisfaction with BC care received (>90%). Specifically, P80+ reported higher overall satisfaction with BC care than P80− (*p* = 0.0015). P80+ reported similar satisfaction overall with the information and support received from their cancer care teams compared to P80− (0.9826).

Satisfaction with BC Information and Support Received Regarding Symptoms/Side Effects of Treatment

In the first year after BC diagnosis, patients of all ages were least satisfied with the information provided regarding (i) the side effects of immunotherapy, (ii) the side effects of targeted biologic therapies, (iii) sexual dysfunction, (iv) the long-term side effects of chemotherapy, and (v) eyebrow/eyelash thinning. 

When compared with P80−, P80+ were significantly less satisfied with information they received regarding (i) the potential side effects of immunotherapy (*p* = 0.0320), targeted biologic therapies (*p* = 0.008), sexual dysfunction (*p* < 0.0001), peripheral neuropathy (*p* < 0.0001), hair loss/thinning (*p* = 0.0030), lymphedema (*p* < 0.0001), hot flashes (*p* < 0.0001), the short and long-term side effects of radiotherapy (*p* < 0.0001 and 0.0277, respectively) and the short term side effects of chemotherapy (*p* < 0.001).

Satisfaction with BC Support Received Regarding Concerns

In the first year after BC diagnosis, patients of all ages reported high satisfaction overall with support provided (~90%). Specifically, there were no significant differences between P80+ and P80− regarding satisfaction with the availability of information pertaining to their BC diagnosis, access to their care team, and the amount of information available regarding their treatment plan. However, P80+ reported lower satisfaction with information provided regarding genetic testing (self) (*p* = 0.0239).

Ranking of Quality Improvement (QI) projects

Compared to P80−, P80+ similarly prioritized proposed QI interventions focusing on (i) the provision of lifetime access to online educational resources (*p* = 0.2588). QI interventions focused on (i) wellness for early breast cancer (EBC) and metastatic breast cancer (MBC) patients and (ii) educational, practical, emotional and holistic support for patients with MBC were also considered high priority but did not differ across age.

Integrative Medicine

P80+ were less likely to have visited an integrative medicine provider for BC treatment than P80− (*p* = 0.0003). P80+ who met with integrative medicine were as likely to have taken supplements or vitamins recommended as P80− (*p* = 0.0056). Despite lower care satisfaction reported overall compared with P80− (*p* = 0.0002), 80% of P80+ were satisfied/very satisfied with care received from integrative medicine. However, data from this section should be cautiously interpreted given the small number of P80+ who responded to these questions.

Medical Second Opinions and Clinical Trial Participation

P80+ were less likely to have received a second opinion (*p* < 0.0001). P80+ were also more likely to have received assistance from a patient advocate regarding BC-related logistics and treatment decisions (*p* = 0.0029).

Regarding clinical trials, 40.5% of P80+ reported prior study participation vs. 47.2% of P80− (*p* < 0.001), and P80+ were less open to participating in a trial in the future (*p* = 0.0001) compared with P80− (18.6% responded “No” and 40.7% responded unsure).

Qualitative Data: Suggestions to Improve BC Care and Prioritization for BC Research

Overall, 77 of 202 P80+ answered the open-ended survey questions (Table 3).

The major themes identified were (1) BC Education, (2) Side Effects of BC Treatment, and (3) Emotional Support During and After BC Diagnosis. Regarding BC education, suggestions included avoiding multiple printed resources unrelated to the patient’s diagnosis and treatment plan, instead providing tailored information based on patient preferences (more vs. less information). Patients also advised that clinicians use less medical terminology and provide thorough information regarding all treatment options. Regarding managing the side effects of BC treatment, suggestions included more detailed information regarding the short and long-term sequelae of local and systemic therapy. Regarding emotional support, patients noted the importance of ongoing access to BC and community support groups. The need for provider education and ongoing efforts to optimize emotional support for BC patients was also highlighted. Other themes included (4) Care Concerns/Provider Sensitivity (need to consider patient frailty, vulnerability and increased susceptibility to discomfort during medical procedures), (5) Survivorship/Long Term Sequelae of BC treatment (advice about surveillance, prosthesis selection, etc.), (6) Diversity, Ethnicity and Inclusion (questions regarding the applicability of BC data for ethnic minorities), (7) BC Care Access (faster surgery and test results), (8) Dense Breast Tissue Screening (insurance coverage), (9) Diet and Exercise (nutrition consultations) and (10) Sexual Dysfunction. Regarding thoughts on what areas of research should be championed, respondents prioritized focusing on efforts to determine the causes of BC, as well as to prevent and cure the same.

## 4. Discussion

We present the results of a large patient experience survey completed by ~2500 BC patients, of whom >200 were P80+; a comparison of both groups shows that P80+ appear to be a distinct population with regard to demographics, treatment-related symptoms, and unmet educational needs. Differences in engagement with integrative medicine and clinical trial participation, as well as the likelihood of pursuing a medical second opinion, were also noted between both groups. Based on our findings, ways to improve BC care in P80+ include: (i) refining patient education materials and (ii) identifying and reducing unique barriers to research participation. On review of the qualitative data, the need for ongoing emotional support during and after BC treatment was also found to be a priority for this population. 

A strength of this substudy is the number of P80+ who responded to the survey, as well as the qualitative data provided from P80+; the latter helped identify relevant themes in this understudied population. Other strengths include the high survey response rate (35%) and the inclusion of rural-dwelling BC patients (15.3% of respondents were P80+), as these patients are often underrepresented in BC care delivery research. Notably, our definition of rurality was strict (RUCA code of 10) and did not include small towns (RUCA codes 7–9 inclusive). Regarding study limitations, most survey respondents had stage 0–3 BC (n = 196; 97%); therefore, concerns germane to P80+ with MBC are not well evaluated. Furthermore, as most survey respondents were married/widowed and Caucasian Christians living in the Midwest of the US, our conclusions may not be generalizable. Therefore, a future study should enroll a more ethnically, racially, and geographically diverse cohort of P80+. Although our survey response rate of 35% was higher than observed in some studies, the substantial number of non-responders could also have contributed to response bias. The higher rate of clinical trial participation reported in our study could also have been a function of response bias, as people who participate in clinical trials are likely more willing to complete a survey. In addition, 155 (76.7%) of the P80+ survey respondents were diagnosed 5–10 years ago, which may have led to recall bias when reporting BC-related symptom severity and concerns. Nevertheless, the inclusion of these patients is important to capture the unique perspectives of P80+ survivors.

P80+ were less likely than younger patients to report that they had been advised to pursue all types of BC therapy. However, when they recalled that this had been recommended, these older patients were as likely as P80− to report that they did indeed receive surgery, radiation therapy and/or chemotherapy. Conversely, if they reported that they had been recommended to receive endocrine therapy or immunotherapy, P80+ were less likely than P80− to report that they had indeed received that treatment. The reasons for therapy decline in older patients may include the duration of therapy, perceived unclear benefit of therapy, concerns regarding the potential for therapies to exacerbate preexisting medical conditions and/or cause new problems, financial concerns, and communication barriers (e.g., hearing or visual impairments that make it difficult to understand the benefits of a given cancer treatment) [12,13]. While the avoidance of potentially toxic therapies is often appropriate in P80+ as risks may outweigh benefits, some older BC patients do benefit from adjuvant chemotherapy [14,15]. The ADVANCE (ADjuVANt Chemotherapy in the Elderly) trial assessed the feasibility of two (neo)adjuvant chemotherapy regimens in parallel-enrolling cohorts of older persons (≥70 years) with human epidermal growth factor receptor 2 (HER2)-negative BC [16]. Although neither regimen met target feasibility goals, trials such as ADVANCE inform the development of tolerable, evidence-based regimens for older BC patients who need chemotherapy. Better age-specific data and personalized decision tools are needed to assist with provider and patient decision-making [17,18,19], as well as the greater integration of existing tools into oncology clinical practice [20,21,22].

From an educational perspective, the level of satisfaction regarding information and support provided to address BC therapy side effects was lower in P80+ than in P80−. Specifically, P80+ were significantly less satisfied with information regarding the potential side effects of immunotherapy, targeted biologic therapies, the short and long-term side effects of radiotherapy, and the short and long-term side effects of chemotherapy. As these topics encompass much of what is discussed at the initial BC consultation(s), the provision of improved educational resources tailored to older patients (e.g., larger print materials and PC use if internet access is available) with additional concessions (e.g., longer visit times and nurse navigator assistance) may improve comprehension and retention of information. Furthermore, a short interval follow-up visit with a care team representative may aid patient decision-making and decrease overwhelming feelings. Even though P80+ were less likely to request a second opinion, those that obtained the same reported a similar level of satisfaction to P80−. A second consultation with an oncology provider provides additional education and may reinforce initial recommendations and improve comprehension [23,24]. A “refresher” visit with a care team representative shortly after the initial consultation to provide a reminder regarding accessing suitable educational materials/resources could be helpful. As P80+ in our substudy also reported less satisfaction with information/education regarding sexual dysfunction, peripheral neuropathy, hair loss/thinning, lymphedema, and several other topics compared with P80−, an institutional pilot study focusing on the delivery of age-appropriate, concise education for P80+ could be implemented. An initial approach could be to focus on one side effect, with patient satisfaction levels assessed thereafter. Given our aging population, strategies focusing on catering to the educational needs of elderly BC patients will increasingly need to be prioritized [25,26].

With regard to the main concerns experienced during the first year after a BC diagnosis, although P80+ reported less anxiety and symptom-related distress compared with P80−, their primary concerns were similar (i.e., emotional distress and anxiety related to the BC diagnosis). Despite the lower levels of emotional distress experienced overall, ongoing access to psychosocial support (in person or virtual) was valued. The importance of “wellness” was stressed, with physical fitness reported as a leading concern. Although P80+ were less inclined to visit an integrative medicine provider, the majority who did were satisfied with their experience; however, these results are based on a relatively small sample size. Given the many benefits of integrative medicine regarding holistic health and pain and symptom management (including for cancer patients) [26,27], efforts to educate older patients and foster connections with licensed professionals should be encouraged.

With regard to clinical trial participation, 60 (40.5%) P80+ reported that they had already participated in a trial (a higher proportion than the general population), but these older patients were significantly less open to doing so in the future compared with P80− (16 [18.6%] stated no and 35 [40.7%] were unsure). Unfortunately, the percentage of P80+ who have participated in/are open to participating in clinical trials is considerably lower in underserved and/or ethnically diverse patients [28,29]. Initiatives to prioritize the inclusion of older persons in research are urgently needed.

P80+ and P80− were well-aligned in recommending the implementation of the following QI initiatives: (i) the provision of lifetime access to online concise patient educational resources; (ii) educational, practical, emotional and holistic support programs for MBC patients; and (iii) BC Wellness Programs for EBC and MBC patients. This implies that, irrespective of age, patients with BC prioritize concise educational resources, continuity of care, ongoing psychological support, and holistic care.

## 5. Conclusions

Efforts to refine care and to address unique challenges in older patients must be a clinical and research priority. Strategies focusing on tailoring educational materials to BC patients diagnosed in their 80s and 90s should be prioritized, as should initiatives that aim to increase clinical trial accrual and promote psychological well-being for P80+. In the future, focused interviews with P80+ could inform interventions to enhance continuity of care, communication, holistic care, and long-term psychosocial support.

## Figures and Tables

**Figure 1 cancers-16-02494-f001:**
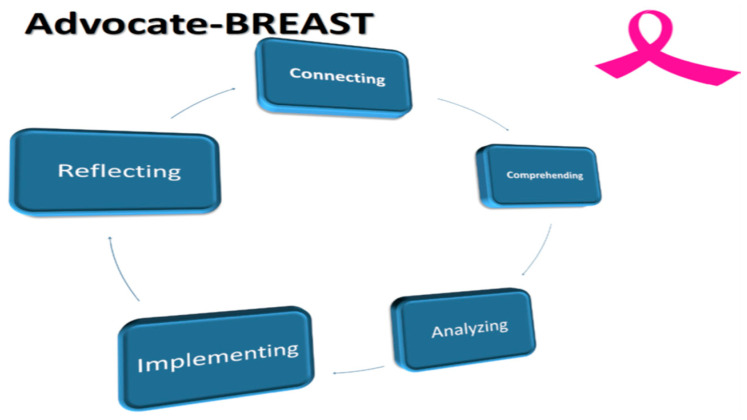
The Advocate-BREAST Project. Legend: The Advocate BREAST project involves 5 distinct steps as follows: 1. Connecting with advocates and patients to determine scope and focus of projects; 2. Comprehending the lived experience of EBC and MBC patients; 3. Developing, circulating surveys and analyzing results in partnership with patient advocates; 4. Implementing results to promote patient centered BC research and treatment; 5.Reflecting on process and considering ways to improve and repeat cycle.

**Figure 2 cancers-16-02494-f002:**
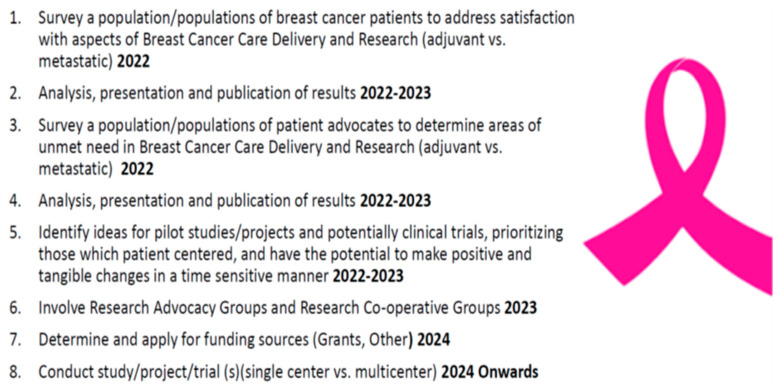
Goals of the Advocate BREAST Project.

**Table 1 cancers-16-02494-t001:** Demographic Information.

	Current Age			
	Under 80 (N = 2235)	80 and Older (N = 202)	Total (N = 2437)	*p*-Value
Demographic Characteristics				
Race, n (%)				0.0136 ^1^
Non-White	65 (2.9%)	0 (0.0%)	65 (2.7%)	
White	2139 (97.1%)	201 (100.0%)	2340 (97.3%)	
Missing	31	1	32	

Ethnicity, n (%)				0.7161 ^1^
Hispanic or Latino	27 (1.3%)	3 (1.6%)	30 (1.3%)	
Not Hispanic or Latino	2125 (98.7%)	189 (98.4%)	2314 (98.7%)	
Missing	83	10	93	

LGBTQIA, n (%)				0.0795 ^1^
Yes	30 (1.4%)	0 (0.0%)	30 (1.3%)	
No	2166 (98.3%)	192 (99.0%)	2358 (98.4%)	
Unsure	7 (0.3%)	2 (1.0%)	9 (0.4%)	
Missing	32	8	40	

Marital status, n (%)				<0.0001 ^1^
Married/living with someone	1811 (81.6%)	121 (61.1%)	1932 (79.9%)	
Separated/divorced	174 (7.8%)	9 (4.5%)	183 (7.6%)	
Widowed	126 (5.7%)	54 (27.3%)	180 (7.4%)	
Never been married	108 (4.9%)	14 (7.1%)	122 (5.0%)	
Missing	16	4	20	

Religion, n (%)				<0.0001 ^1^
Catholic	630 (28.5%)	50 (25.1%)	680 (28.2%)	
Jewish/Ashkenazi Jewish	23 (1.0%)	2 (1.0%)	25 (1.0%)	
Protestant	830 (37.5%)	115 (57.8%)	945 (39.2%)	
Islam/Muslim	37 (1.7%)	7 (3.5%)	44 (1.8%)	
Other	392 (17.7%)	13 (6.5%)	405 (16.8%)	
None	302 (13.6%)	12 (6.0%)	314 (13.0%)	
Missing	21	3	24	

English is first language, n (%)				0.1874 ^1^
Yes	2145 (97.5%)	196 (99.0%)	2341 (97.6%)	
No	55 (2.5%)	2 (1.0%)	57 (2.4%)	
Missing	35	4	39	

Residency based on Census tract, n (%)				0.0136 ^1^
Non-Rural	2008 (90.2%)	171 (84.7%)	2179 (89.7%)	
Rural	219 (9.8%)	31 (15.3%)	250 (10.3%)	
Missing	8	0	8	

Patient has primary care provider at Mayo Clinic, n (%)				0.3645 ^1^
No	1465 (65.5%)	126 (62.4%)	1591 (65.3%)	
Yes	770 (34.5%)	76 (37.6%)	846 (34.7%)	

Months since breast cancer diagnosis, n (%)				<0.0001 ^1^
<12 months	181 (8.1%)	4 (2.0%)	185 (7.6%)	
12–23 months	183 (8.2%)	6 (3.0%)	189 (7.8%)	
24–35 months	222 (9.9%)	7 (3.5%)	229 (9.4%)	
36–59 months	397 (17.8%)	30 (14.9%)	427 (17.5%)	
60–119 months	530 (23.7%)	43 (21.3%)	573 (23.5%)	
120+ months	722 (32.3%)	112 (55.4%)	834 (34.2%)	

BC stage, n (%)				0.2571 ^1^
Stage 0	361 (17.7%)	31 (15.7%)	392 (17.5%)	
Stage 1–3	1640 (80.4%)	165 (83.8%)	1805 (80.7%)	
Stage 4	40 (2.0%)	1 (0.5%)	41 (1.8%)	
Missing	194	5	199	

Primary Breast Cancer Treatment Location(s), n (%)				0.0531 ^1^
Mayo Clinic Sites	1528 (68.8%)	152 (76.8%)	1680 (69.4%)	
Other (Non-Mayo Clinic)	135 (6.1%)	7 (3.5%)	142 (5.9%)	
Both (Mayo Clinic and Non-Mayo Clinic)	559 (25.2%)	39 (19.7%)	598 (24.7%)	
Missing	13	4	17	

Was surgery recommended, n (%)?				0.0062 ^1^
Yes	2135 (96.9%)	181 (92.8%)	2316 (96.6%)	
No	58 (2.6%)	13 (6.7%)	71 (3.0%)	
Unsure	10 (0.5%)	1 (0.5%)	11 (0.5%)	
Missing	32	7	39	

If surgery was recommended, did you proceed with it, n (%)?				0.5643 ^1^
Yes	2075 (98.6%)	175 (97.8%)	2250 (98.5%)	
No	28 (1.3%)	4 (2.2%)	32 (1.4%)	
Unsure	2 (0.1%)	0 (0.0%)	2 (0.1%)	
Missing	130	23	153	

Was chemotherapy recommended, n (%)?				<0.0001 ^1^
Yes	950 (43.2%)	42 (21.8%)	992 (41.5%)	
No	1221 (55.6%)	142 (73.6%)	1363 (57.0%)	
Unsure	27 (1.2%)	9 (4.7%)	36 (1.5%)	
Missing	37	9	46	

If chemotherapy was recommended, did you proceed with it, n (%)?				0.1641 ^1^
Yes	878 (93.4%)	36 (87.8%)	914 (93.2%)	
No	62 (6.6%)	5 (12.2%)	67 (6.8%)	
Missing	1295	161	1456	

Was radiotherapy recommended, n (%)?				0.0037 ^1^
Yes	1463 (66.5%)	111 (58.4%)	1574 (65.9%)	
No	705 (32.0%)	71 (37.4%)	776 (32.5%)	
Unsure	32 (1.5%)	8 (4.2%)	40 (1.7%)	
Missing	35	12	47	

If radiotherapy was recommended, did you proceed with it, n (%)?				0.6635 ^1^
Yes	1346 (93.9%)	102 (92.7%)	1448 (93.8%)	
No	82 (5.7%)	8 (7.3%)	90 (5.8%)	
Unsure	5 (0.3%)	0 (0.0%)	5 (0.3%)	
Missing	802	92	894	

Was endocrine (anti-estrogen) therapy recommended, n (%)?				<0.0001 ^1^
Yes	1296 (59.2%)	68 (35.8%)	1364 (57.3%)	
No	708 (32.3%)	76 (40.0%)	784 (33.0%)	
Unsure	185 (8.5%)	46 (24.2%)	231 (9.7%)	
Missing	46	12	58	

If endocrine therapy was recommended, did you proceed with it, n (%)?				0.0011 ^1^
Yes	1154 (90.8%)	57 (85.1%)	1211 (90.5%)	
No	114 (9.0%)	8 (11.9%)	122 (9.1%)	
Unsure	3 (0.2%)	2 (3.0%)	5 (0.4%)	
Missing	964	135	1099	

Was targeted biologic therapy recommended, n (%)?				<0.0001 ^1^
Yes	104 (4.8%)	5 (2.7%)	109 (4.6%)	
No	1707 (78.1%)	121 (65.1%)	1828 (77.1%)	
Unsure	374 (17.1%)	60 (32.3%)	434 (18.3%)	
Missing	50	16	66	

If targeted biologic therapy was recommended, did you proceed with it, n (%)?				0.0940 ^1^
Yes	99 (96.1%)	4 (80.0%)	103 (95.4%)	
No	4 (3.9%)	1 (20.0%)	5 (4.6%)	
Missing	2132	197	2329	

Was immunotherapy recommended, n (%)?				<0.0001 ^1^
Yes	100 (4.6%)	3 (1.6%)	103 (4.3%)	
No	1786 (81.3%)	138 (73.0%)	1924 (80.6%)	
Unsure	311 (14.2%)	48 (25.4%)	359 (15.0%)	
Missing	38	13	51	

If immunotherapy was recommended, did you proceed with it, n (%)?				0.0015 ^1^
Yes	98 (98.0%)	2 (66.7%)	100 (97.1%)	
No	2 (2.0%)	1 (33.3%)	3 (2.9%)	
Missing	2135	199	2334	


^1^ Chi-Square *p*-value.

**Table 2 cancers-16-02494-t002:** Survey Responses.

	Unweighted			Weighted		
	Under 80 (N = 2235)	80 and Older (N = 202)	*p*–Value	Under 80	80 and Older	*p*-Value
	Level of symptom experienced in first year after breast cancer diagnosis (10-point scale, 0 = none, 10 = as bad as I could imagine)					

Hair loss/thinning			<0.0001 ^2^			0.0412 ^4^
N	2180	191				
Mean (SD)	4.0 (4.23)	2.2 (3.41)		3.9 (3.21)	3.5 (9.37)	
Median (IQR)	2.0 (0.0, 9.0)	0.0 (0.0, 4.0)		2.0 (0.0–9.0)	2.0 (0.0–6.0)	

Hot flashes			<0.0001 ^2^			<0.0001 ^4^
N	2179	190				
Mean (SD)	3.8 (3.52)	1.5 (2.51)		3.8 (2.68)	1.6 (6.12)	
Median (IQR)	3.0 (0.0, 7.0)	0.0 (0.0, 2.0)		3.0 (0.0–7.0)	0.0 (0.0–2.0)	

Eyebrow/Eyelash thinning			<0.0001 ^2^			<0.0001 ^4^
N	2164	189				
Mean (SD)	3.6 (4.11)	1.6 (2.98)		3.5 (3.11)	2.2 (7.74)	
Median (IQR)	1.0 (0.0, 8.0)	0.0 (0.0, 2.0)		1.0 (0.0–8.0)	0.0 (0.0–4.0)	

Sexual dysfunction			<0.0001 ^2^			<0.0001 ^4^
N	2154	184				
Mean (SD)	3.2 (3.36)	0.9 (2.10)		3.2 (2.55)	1.0 (5.54)	
Median (IQR)	2.0 (0.0, 6.0)	0.0 (0.0, 0.0)		2.0 (0.0–6.0)	0.0 (0.0–0.0)	

Cognitive/memory issues (i.e., brain fog or chemo brain)			<0.0001 ^2^			<0.0001 ^4^
N	2180	192				
Mean (SD)	2.9 (3.25)	0.8 (1.73)		2.8 (2.47)	1.0 (4.46)	
Median (IQR)	2.0 (0.0, 6.0)	0.0 (0.0, 1.0)		1.0 (0.0–5.0)	0.0 (0.0–2.0)	

The numbness and/or tingling in your hands and/or feet			<0.0001 ^2^			0.9199 ^4^
N	2180	190				
Mean (SD)	2.1 (2.91)	1.2 (2.39)		2.0 (2.21)	2.0 (7.04)	
Median (IQR)	0.0 (0.0, 3.0)	0.0 (0.0, 1.0)		0.0 (0.0–3.0)	0.0 (0.0–3.0)	

The swelling in your arm or arms			0.0137 ^2^			<0.0001 ^4^
N	2176	191				
Mean (SD)	1.2 (2.24)	0.8 (1.84)		1.2 (1.71)	0.8 (4.28)	
Median (IQR)	0.0 (0.0, 1.0)	0.0 (0.0, 0.0)		0.0 (0.0–1.0)	0.0 (0.0–0.0)	

	Level of concern regarding issues within first year after breast cancer diagnosis (10-point scale, 0 = not at all concerned, 10 = highly concerned)					

Breast cancer recurrence or spread			<0.0001 ^2^			<0.0001 ^4^
N	2161	185				
Mean (SD)	4.9 (3.62)	2.7 (3.16)		4.8 (2.78)	3.3 (8.11)	
Median (IQR)	5.0 (1.0, 8.0)	2.0 (0.0, 5.0)		5.0 (1.0–8.0)	2.0 (0.0–5.0)	

Concerns as regards how loved ones will cope practically and emotionally if you pass away from breast cancer			<0.0001 ^2^			<0.0001 ^4^
N	2159	180				
Mean (SD)	4.7 (3.68)	2.7 (2.98)		4.6 (2.81)	3.6 (8.43)	
Median (IQR)	5.0 (1.0, 8.0)	2.0 (0.0, 5.0)		5.0 (1.0–8.0)	2.0 (0.0–6.0)	

Your diagnosis and prognosis			<0.0001 ^2^			<0.0001 ^4^
N	2158	181				
Mean (SD)	4.4 (3.39)	2.5 (3.02)		4.4 (2.58)	3.2 (8.21)	
Median (IQR)	4.0 (1.0, 8.0)	1.0 (0.0, 4.0)		4.0 (1.0–7.0)	2.0 (0.0–5.0)	

Fear of dying from breast cancer			<0.0001 ^2^			<0.0001 ^4^
N	2166	184				
Mean (SD)	4.0 (3.53)	2.0 (2.61)		4.0 (2.68)	2.5 (7.17)	
Median (IQR)	3.0 (1.0, 7.0)	1.0 (0.0, 3.0)		3.0 (1.0–7.0)	1.0 (0.0–5.0)	

Your weight and/or physical fitness level			<0.0001 ^2^			<0.0001 ^4^
N	2163	185				
Mean (SD)	3.9 (3.12)	1.9 (2.42)		3.9 (2.37)	2.4 (6.28)	
Median (IQR)	4.0 (1.0, 6.0)	1.0 (0.0, 4.0)		4.0 (1.0–6.0)	2.0 (0.0–4.0)	

Your emotional health			<0.0001 ^2^			<0.0001 ^4^
N	2149	182				
Mean (SD)	4.0 (3.19)	1.7 (2.42)		3.9 (2.43)	2.0 (6.17)	
Median (IQR)	4.0 (1.0, 7.0)	1.0 (0.0, 3.0)		3.0 (1.0–7.0)	1.0 (0.0–3.0)	

The emotional health of others (partner, children, other family members)			<0.0001 ^2^			<0.0001 ^4^
N	2161	185				
Mean (SD)	3.5 (3.17)	1.5 (2.26)		3.5 (2.42)	2.0 (6.32)	
Median (IQR)	3.0 (0.0, 6.0)	0.0 (0.0, 2.0)		3.0 (0.0–6.0)	1.0 (0.0–3.0)	

Your quality of life			<0.0001 ^2^			<0.0001 ^4^
N	2158	182				
Mean (SD)	3.5 (3.18)	1.9 (2.73)		3.4 (2.42)	2.3 (6.94)	
Median (IQR)	3.0 (0.0, 6.0)	0.0 (0.0, 3.0)		3.0 (0.0–6.0)	1.0 (0.0–4.0)	

Your bone health			0.0001 ^2^			<0.0001 ^4^
N	2163	183				
Mean (SD)	3.4 (3.22)	2.5 (2.90)		3.4 (2.44)	2.6 (6.66)	
Median (IQR)	3.0 (0.0, 6.0)	1.0 (0.0, 5.0)		3.0 (0.0–6.0)	2.0 (0.0–5.0)	

Your nutrition/diet			<0.0001 ^2^			<0.0001 ^4^
N	2159	183				
Mean (SD)	3.3 (3.07)	1.7 (2.42)		3.3 (2.33)	2.0 (5.96)	
Median (IQR)	3.0 (0.0, 5.0)	0.0 (0.0, 3.0)		3.0 (0.0–5.0)	1.0 (0.0–4.0)	

Your self-esteem/self-confidence			<0.0001 ^2^			<0.0001 ^4^
N	2157	185				
Mean (SD)	3.3 (3.17)	1.6 (2.43)		3.3 (2.42)	1.8 (6.32)	
Median (IQR)	2.0 (0.0, 6.0)	0.0 (0.0, 2.0)		2.0 (0.0–6.0)	0.0 (0.0–3.0)	

Changes in intimacy with your partner			<0.0001 ^2^			<0.0001 ^4^
N	2134	183				
Mean (SD)	3.4 (3.50)	0.8 (1.96)		3.3 (2.67)	1.0 (5.69)	
Median (IQR)	2.0 (0.0, 6.0)	0.0 (0.0, 0.0)		2.0 (0.0–6.0)	0.0 (0.0–0.0)	

Body image/attractiveness/sexuality			<0.0001 ^2^			<0.0001 ^4^
N	2156	184				
Mean (SD)	3.4 (3.37)	1.1 (2.10)		3.3 (2.57)	1.5 (5.82)	
Median (IQR)	2.0 (0.0, 6.0)	0.0 (0.0, 2.0)		2.0 (0.0–6.0)	0.0 (0.0–2.0)	

Existential Issues (anxiety and/or questions regarding the meaning of life and/or your purpose in life)			<0.0001 ^2^			<0.0001 ^4^
N	2155	183				
Mean (SD)	3.0 (3.25)	1.5 (2.33)		3.0 (2.48)	1.8 (6.17)	
Median (IQR)	2.0 (0.0, 5.0)	0.0 (0.0, 2.0)		2.0 (0.0–5.0)	0.0 (0.0–3.0)	

Pressure to keep positive about your breast cancer diagnosis			<0.0001 ^2^			<0.0001 ^4^
N	2157	183				
Mean (SD)	2.7 (3.18)	1.3 (2.35)		2.7 (2.42)	1.8 (6.82)	
Median (IQR)	1.0 (0.0, 5.0)	0.0 (0.0, 1.0)		1.0 (0.0–5.0)	0.0 (0.0–3.0)	

Your heart health			0.0005 ^2^			0.0395 ^4^
N	2161	186				
Mean (SD)	2.5 (3.00)	1.8 (2.62)		2.5 (2.28)	2.2 (7.20)	
Median (IQR)	1.0 (0.0, 5.0)	0.0 (0.0, 3.0)		1.0 (0.0–5.0)	0.0 (0.0–5.0)	

Genetic testing/counseling for family members			<0.0001 ^2^			<0.0001 ^4^
N	2143	182				
Mean (SD)	2.6 (3.27)	1.3 (2.49)		2.5 (2.48)	1.9 (6.47)	
Median (IQR)	1.0 (0.0, 5.0)	0.0 (0.0, 2.0)		1.0 (0.0–5.0)	0.0 (0.0–3.0)	

Advocating for yourself as it relates to your breast cancer diagnosis			<0.0001 ^2^			<0.0001 ^4^
N	2141	184				
Mean (SD)	2.6 (3.24)	1.1 (2.21)		2.6 (2.47)	1.5 (5.78)	
Median (IQR)	1.0 (0.0, 5.0)	0.0 (0.0, 1.0)		1.0 (0.0–5.0)	0.0 (0.0–2.0)	

Genetic testing/counseling for yourself			<0.0001 ^2^			<0.0001 ^4^
N	2143	181				
Mean (SD)	2.5 (3.18)	1.1 (2.40)		2.4 (2.41)	1.5 (6.51)	
Median (IQR)	1.0 (0.0, 5.0)	0.0 (0.0, 1.0)		1.0 (0.0–5.0)	0.0 (0.0–2.0)	

The impact of your cancer diagnosis on your home responsibilities			<0.0001 ^2^			<0.0001 ^4^
N	2158	181				
Mean (SD)	2.5 (2.98)	1.1 (2.06)		2.4 (2.26)	1.5 (6.18)	
Median (IQR)	1.0 (0.0, 5.0)	0.0 (0.0, 1.0)		1.0 (0.0–4.0)	0.0 (0.0–2.0)	

Changes in your relationships with friends and/or family members			<0.0001 ^2^			<0.0001 ^4^
N	2153	185				
Mean (SD)	2.3 (2.94)	0.7 (1.83)		2.2 (2.22)	1.2 (5.98)	
Median (IQR)	1.0 (0.0, 4.0)	0.0 (0.0, 0.0)		1.0 (0.0–4.0)	0.0 (0.0–1.0)	

Ease of access to/communication with your oncology care team			0.0229 ^2^			0.0475 ^4^
N	2154	182				
Mean (SD)	1.9 (2.95)	1.4 (2.71)		1.9 (2.25)	1.7 (6.78)	
Median (IQR)	0.0 (0.0, 3.0)	0.0 (0.0, 1.0)		0.0 (0.0–3.0)	0.0 (0.0–2.0)	

The impact of your cancer diagnosis on your employment status and your career			<0.0001 ^2^			<0.0001 ^4^
N	2153	183				
Mean (SD)	1.9 (3.04)	0.5 (1.99)		1.9 (2.29)	0.4 (3.78)	
Median (IQR)	0.0 (0.0, 3.0)	0.0 (0.0, 0.0)		0.0 (0.0–3.0)	0.0 (0.0–0.0)	

Pressure to keep family and friends updated as regards your breast cancer diagnosis and the plan for treatment			<0.0001 ^2^			<0.0001 ^4^
N	2150	182				
Mean (SD)	1.9 (2.57)	0.9 (2.09)		1.8 (1.96)	1.3 (6.22)	
Median (IQR)	1.0 (0.0, 3.0)	0.0 (0.0, 1.0)		1.0 (0.0–3.0)	0.0 (0.0–1.0)	

The need for privacy regarding your breast cancer diagnosis			<0.0001 ^2^			<0.0001 ^4^
N	2150	183				
Mean (SD)	1.8 (2.78)	1.0 (2.26)		1.8 (2.11)	1.1 (5.61)	
Median (IQR)	0.0 (0.0, 3.0)	0.0 (0.0, 0.0)		0.0 (0.0–3.0)	0.0 (0.0–1.0)	

The impact of your breast cancer diagnosis on dating and/or socializing			<0.0001 ^2^			<0.0001 ^4^
N	2146	184				
Mean (SD)	1.6 (2.72)	0.4 (1.51)		1.6 (2.07)	0.8 (4.26)	
Median (IQR)	0.0 (0.0, 2.0)	0.0 (0.0, 0.0)		0.0 (0.0–2.0)	0.0 (0.0–0.0)	

Your fertility			0.0012 ^2^			<0.0001 ^4^
N	2147	182				
Mean (SD)	0.5 (2.01)	0.0 (0.35)		0.5 (1.52)	0.0 (0.92)	
Median (IQR)	0.0 (0.0, 0.0)	0.0 (0.0, 0.0)		0.0 (0.0–0.0)	0.0 (0.0–0.0)	

Cultural and/or religious concerns related to your breast cancer diagnosis and treatment			0.9544 ^2^			0.6661 ^4^
N	2149	184				
Mean (SD)	0.4 (1.25)	0.4 (1.41)		0.4 (0.94)	0.4 (3.34)	
Median (IQR)	0.0 (0.0, 0.0)	0.0 (0.0, 0.0)		0.0 (0.0–0.0)	0.0 (0.0–0.0)	

	Satisfaction of breast cancer patient experience (10-point scale, 0 = very dissatisfied, 10 = very satisfied)					

How satisfied have you been with your breast cancer care overall to date			0.1566 ^2^			0.0015 ^4^
N	2155	183				
Mean (SD)	8.9 (1.76)	9.1 (2.07)		8.9 (1.35)	9.1 (4.78)	
Median (IQR)	10.0 (8.0, 10.0)	10.0 (9.0, 10.0)		10 (8.0–10)	10 (9.0–10)	

How satisfied were/are you overall with the information and support you receive(d) from your cancer care teams			0.6385 ^2^			0.9826 ^4^
N	2159	183				
Mean (SD)	8.9 (1.83)	8.8 (2.46)		8.9 (1.41)	8.9 (5.91)	
Median (IQR)	10.0 (8.0, 10.0)	10.0 (9.0, 10.0)		10 (8.0–10)	10 (9.0–10)	

	Satisfaction of breast cancer information and support received from cancer care team for symptoms (10-point scale, 0 = very dissatisfied, 10 = very satisfied)					

The short-term side effects of breast cancer surgery			0.0140 ^2^			0.5266 ^4^
N	1844	119				
Mean (SD)	7.8 (2.77)	7.2 (3.41)		7.8 (2.12)	7.7 (8.27)	
Median (IQR)	9.0 (7.0, 10.0)	9.0 (5.0, 10.0)		9.0 (7.0–10)	9.0 (6.0–10)	

The short-term side effects of radiotherapy			<0.0001 ^2^			<0.0001 ^4^
N	1318	82				
Mean (SD)	7.7 (2.99)	6.3 (3.89)		7.6 (2.31)	6.8 (9.97)	
Median (IQR)	9.0 (7.0, 10.0)	8.0 (2.0, 10.0)		9.0 (6.0–10)	8.0 (5.0–10)	

The long-term side effects of breast cancer surgery			0.0730 ^2^			0.7805 ^4^
N	1825	118				
Mean (SD)	7.3 (2.97)	6.8 (3.63)		7.3 (2.28)	7.3 (8.81)	
Median (IQR)	8.0 (5.0, 10.0)	8.5 (4.0, 10.0)		8.0 (5.0–10)	9.0 (5.0–10)	

The side effects of endocrine therapy (i.e., hormone blocking treatment)			0.1811 ^2^			0.0638 ^4^
N	1256	64				
Mean (SD)	7.0 (3.09)	6.4 (3.75)		6.9 (2.35)	7.3 (8.61)	
Median (IQR)	8.0 (5.0, 10.0)	8.0 (4.0, 10.0)		8.0 (5.0–10)	9.0 (5.0–10)	

The short-term side effects of chemotherapy			<0.0001 ^2^			<0.0001 ^4^
N	942	49				
Mean (SD)	7.1 (3.30)	4.2 (3.99)		7.0 (2.51)	5.5 (10.5)	
Median (IQR)	8.0 (5.0, 10.0)	4.0 (0.0, 8.0)		8.0 (5.0–10)	6.0 (2.0–8.0)	

The long-term side effects of radiotherapy			0.0277 ^2^			<0.0001 ^4^
N	1287	76				
Mean (SD)	7.0 (3.24)	6.1 (3.87)		6.9 (2.47)	6.0 (9.99)	
Median (IQR)	8.0 (5.0, 10.0)	8.0 (2.5, 10.0)		8.0 (5.0–10)	8.0 (4.0–9.0)	

Hot flashes			<0.0001 ^2^			<0.0001 ^4^
N	1347	63				
Mean (SD)	7.0 (3.05)	4.8 (4.22)		7.0 (2.32)	4.5 (9.22)	
Median (IQR)	8.0 (5.0, 10.0)	5.0 (0.0, 10.0)		8.0 (5.0–10)	4.0 (0.0–10)	

The swelling in your arm or arms			<0.0001 ^2^			<0.0001 ^4^
N	933	67				
Mean (SD)	6.7 (3.64)	4.6 (4.24)		6.6 (2.78)	4.4 (11.1)	
Median (IQR)	8.0 (4.0, 10.0)	5.0 (0.0, 9.0)		8.0 (4.0–10)	4.0 (0.0–9.0)	

Hair loss/thinning			0.0030 ^2^			0.0066 ^4^
N	1214	77				
Mean (SD)	6.5 (3.40)	5.4 (3.74)		6.5 (2.58)	6.0 (9.64)	
Median (IQR)	8.0 (4.0, 10.0)	5.0 (2.0, 9.0)		8.0 (4.0–10)	7.0 (3.0–9.0)	

The numbness and/or tingling in your hands and/or feet			<0.0001 ^2^			0.0062 ^4^
N	1064	70				
Mean (SD)	6.5 (3.39)	4.3 (3.84)		6.4 (2.58)	5.8 (10.2)	
Median (IQR)	8.0 (4.0, 9.0)	4.0 (0.0, 8.0)		8.0 (4.0–9.0)	7.0 (3.0–10)	

Eyebrow/Eyelash thinning			0.0070 ^2^			0.1417 ^4^
N	1107	71				
Mean (SD)	6.3 (3.48)	5.1 (3.84)		6.2 (2.64)	5.9 (10.2)	
Median (IQR)	7.0 (4.0, 10.0)	5.0 (1.0, 9.0)		7.0 (4.0–10)	7.0 (3.0–9.0)	

The long-term side effects of chemotherapy			0.0021 ^2^			0.4551 ^4^
N	932	50				
Mean (SD)	6.2 (3.46)	4.6 (4.03)		6.1 (2.61)	6.0 (10.5)	
Median (IQR)	7.0 (3.0, 9.0)	5.0 (0.0, 8.0)		7.0 (3.0–9.0)	8.0 (3.0–8.0)	

Sexual dysfunction			<0.0001 ^2^			<0.0001 ^4^
N	1125	51				
Mean (SD)	5.8 (3.34)	3.0 (3.69)		5.8 (2.53)	2.7 (7.28)	
Median (IQR)	6.0 (3.0, 9.0)	2.0 (0.0, 5.0)		6.0 (3.0–9.0)	2.0 (0.0–3.0)	

The potential side effects of targeted biologic therapies			0.0008 ^2^			0.0110 ^4^
N	342	22				
Mean (SD)	4.9 (4.23)	1.8 (3.43)		4.8 (3.20)	3.7 (11.1)	
Median (IQR)	5.0 (0.0, 9.0)	0.0 (0.0, 2.0)		5.0 (0.0–9.0)	5.0 (0.0–5.0)	

The side effects of immunotherapy			0.0320 ^2^			0.0293 ^4^
N	343	26				
Mean (SD)	4.4 (4.07)	2.6 (3.91)		4.3 (3.08)	3.3 (9.81)	
Median (IQR)	4.0 (0.0, 9.0)	0.0 (0.0, 5.0)		4.0 (0.0–8.0)	0.0 (0.0–7.0)	

	Satisfaction of breast cancer information and support received from cancer care team for concerns (10-point scale, 0 = very dissatisfied, 10 = very satisfied)					

Availability of information related to your cancer diagnosis (e.g., scan results, medical notes)			0.7445 ^2^			0.0793 ^4^
N	1878	133				
Mean (SD)	8.8 (2.09)	8.9 (2.34)		8.8 (1.61)	9.0 (5.74)	
Median (IQR)	10.0 (8.0, 10.0)	10.0 (9.0, 10.0)		10 (8.0–10)	10 (9.0–10)	

Ease of access to/communication with your cancer care team			0.8561 ^2^			0.3358 ^4^
N	1903	142				
Mean (SD)	8.8 (2.06)	8.7 (2.36)		8.8 (1.59)	8.7 (5.74)	
Median (IQR)	10.0 (8.0, 10.0)	10.0 (8.0, 10.0)		10 (8.0–10)	10 (9.0–10)	

The amount of information available to you about your breast cancer diagnosis and your treatment plan			0.9373 ^2^			0.5537 ^4^
N	1884	139				
Mean (SD)	8.7 (2.17)	8.7 (2.46)		8.7 (1.66)	8.6 (5.73)	
Median (IQR)	10.0 (8.0, 10.0)	10.0 (8.0, 10.0)		10 (8.0–10)	10 (8.0–10)	

Your diagnosis and prognosis			0.2169 ^2^			0.5283 ^4^
N	1843	123				
Mean (SD)	8.1 (2.41)	7.9 (3.02)		8.1 (1.84)	8.2 (6.91)	
Median (IQR)	9.0 (8.0, 10.0)	9.0 (7.0, 10.0)		9.0 (8.0–10)	9.0 (8.0–10)	

Advocating for yourself as it relates to your breast cancer diagnosis			0.6014 ^2^			0.7448 ^4^
N	1502	95				
Mean (SD)	7.9 (2.70)	8.1 (2.84)		7.9 (2.05)	8.0 (6.65)	
Median (IQR)	9.0 (7.0, 10.0)	10.0 (7.0, 10.0)		9.0 (7.0–10)	9.0 (8.0–10)	

Genetic testing/counseling for yourself			0.0239 ^2^			<0.0001 ^4^
N	1334	54				
Mean (SD)	7.9 (2.93)	6.9 (3.41)		7.8 (2.25)	7.0 (7.99)	
Median (IQR)	9.0 (7.0, 10.0)	8.0 (5.0, 10.0)		9.0 (7.0–10)	7.0 (5.0–10)	

Breast cancer recurrence or spread			0.0889 ^2^			0.7230 ^4^
N	1686	111				
Mean (SD)	7.6 (2.64)	7.1 (3.36)		7.6 (2.01)	7.5 (8.16)	
Median (IQR)	8.0 (6.0, 10.0)	8.0 (5.0, 10.0)		8.0 (6.0–10)	9.0 (6.0–10)	

The need for privacy regarding your breast cancer diagnosis			0.1535 ^2^			<0.0001 ^4^
N	1127	62				
Mean (SD)	7.5 (3.11)	6.9 (3.73)		7.5 (2.38)	6.1 (10.1)	
Median (IQR)	9.0 (5.0, 10.0)	8.0 (5.0, 10.0)		9.0 (5.0–10)	7.0 (2.0–10)	

Your quality of life			0.9638 ^2^			0.5924 ^4^
N	1583	94				
Mean (SD)	7.4 (2.74)	7.4 (3.41)		7.4 (2.10)	7.5 (7.90)	
Median (IQR)	8.0 (5.0, 10.0)	9.0 (7.0, 10.0)		8.0 (6.0–10)	8.0 (7.0–10)	

Your bone health			0.8132 ^2^			0.0122 ^4^
N	1522	104				
Mean (SD)	7.3 (2.87)	7.2 (3.05)		7.3 (2.19)	7.7 (7.07)	
Median (IQR)	8.0 (5.0, 10.0)	8.0 (5.0, 10.0)		8.0 (5.0–10)	8.0 (5.0–10)	

Your emotional health			0.0583 ^2^			0.0023 ^4^
N	1620	89				
Mean (SD)	7.2 (2.81)	6.7 (3.46)		7.2 (2.14)	6.8 (7.86)	
Median (IQR)	8.0 (5.0, 10.0)	8.0 (5.0, 10.0)		8.0 (5.0–10)	8.0 (5.0–10)	

Ability to perform home responsibilities			0.3528 ^2^			0.6902 ^4^
N	1276	79				
Mean (SD)	7.2 (2.98)	6.9 (3.67)		7.2 (2.27)	7.3 (7.69)	
Median (IQR)	8.0 (5.0, 10.0)	9.0 (4.0, 10.0)		8.0 (5.0–10)	8.0 (6.0–10)	

Your self-esteem/self-confidence			0.1390 ^2^			0.0012 ^4^
N	1469	97				
Mean (SD)	7.1 (2.91)	7.5 (3.15)		7.1 (2.22)	7.6 (7.27)	
Median (IQR)	8.0 (5.0, 10.0)	9.0 (6.0, 10.0)		8.0 (5.0–10)	8.0 (7.0–10)	

Your weight and/or physical fitness level			0.0790 ^2^			0.0242 ^4^
N	1487	84				
Mean (SD)	7.1 (2.81)	6.5 (3.24)		7.1 (2.15)	6.8 (7.17)	
Median (IQR)	8.0 (5.0, 10.0)	8.0 (5.0, 9.0)		8.0 (5.0–10)	8.0 (5.0–9.0)	

Nutrition/diet			0.3873 ^2^			0.2325 ^4^
N	1499	84				
Mean (SD)	7.1 (2.89)	6.8 (3.35)		7.1 (2.19)	6.9 (7.85)	
Median (IQR)	8.0 (5.0, 10.0)	8.0 (5.0, 10.0)		8.0 (5.0–10)	8.0 (5.0–9.0)	

Fear of dying from breast cancer			0.4561 ^2^			0.8885 ^4^
N	1444	94				
Mean (SD)	7.0 (2.93)	6.8 (3.29)		7.0 (2.24)	7.0 (7.72)	
Median (IQR)	8.0 (5.0, 10.0)	8.0 (5.0, 10.0)		8.0 (5.0–10)	9.0 (5.0–9.0)	

Genetic testing/counseling for family members			0.2136 ^2^			0.0271 ^4^
N	1072	52				
Mean (SD)	7.0 (3.34)	6.4 (3.52)		7.0 (2.55)	6.5 (9.12)	
Median (IQR)	8.0 (5.0, 10.0)	7.0 (4.5, 10.0)		8.0 (5.0–10)	7.0 (5.0–10)	

Amount of information available to you about diet, nutrition and supplements			0.2805 ^2^			0.2207 ^4^
N	1675	102				
Mean (SD)	6.9 (3.09)	7.3 (3.11)		7.0 (2.34)	7.1 (7.35)	
Median (IQR)	8.0 (5.0, 10.0)	8.0 (5.0, 10.0)		8.0 (5.0–10)	8.0 (5.0–10)	

Your heart health			0.7540 ^2^			0.0305 ^4^
N	1355	91				
Mean (SD)	6.9 (2.98)	6.8 (3.16)		7.0 (2.26)	7.3 (7.06)	
Median (IQR)	8.0 (5.0, 10.0)	8.0 (5.0, 10.0)		8.0 (5.0–10)	8.0 (5.0–10)	

Amount of information available to you and your next of kin about advanced care directives, palliative medicine, hospice			0.4259 ^2^			0.8057 ^4^
N	969	70				
Mean (SD)	6.9 (3.34)	7.2 (3.57)		6.9 (2.55)	6.8 (8.38)	
Median (IQR)	8.0 (5.0, 10.0)	9.0 (5.0, 10.0)		8.0 (5.0–10)	7.0 (5.0–10)	

Pressure to keep family and friends updated as regards your breast cancer diagnosis and the plan for treatment			0.2879 ^2^			0.0286 ^4^
N	1086	69				
Mean (SD)	6.9 (3.10)	7.3 (3.57)		6.9 (2.37)	7.3 (8.67)	
Median (IQR)	8.0 (5.0, 10.0)	9.0 (5.0, 10.0)		8.0 (5.0–10)	9.0 (6.0–10)	

Pressure to keep positive about your breast cancer diagnosis			0.6017 ^2^			0.8120 ^4^
N	1182	74				
Mean (SD)	6.7 (3.14)	6.9 (3.61)		6.7 (2.39)	6.8 (8.75)	
Median (IQR)	8.0 (5.0, 10.0)	8.5 (5.0, 10.0)		8.0 (5.0–10)	8.0 (5.0–10)	

The emotional health of others (partner, children, other family members)			0.2264 ^2^			0.8171 ^4^
N	1389	78				
Mean (SD)	6.7 (3.05)	6.3 (3.42)		6.7 (2.32)	6.8 (8.06)	
Median (IQR)	8.0 (5.0, 10.0)	7.0 (4.0, 9.0)		8.0 (5.0–10)	8.0 (5.0–9.0)	

Cultural and/or religious concerns related to your breast cancer diagnosis and treatment			0.8936 ^2^			0.0895 ^4^
N	585	39				
Mean (SD)	6.5 (3.92)	6.4 (4.27)		6.5 (2.99)	7.0 (9.43)	
Median (IQR)	8.0 (3.0, 10.0)	9.0 (0.0, 10.0)		8.0 (3.0–10)	9.0 (6.0–10)	

Amount of information available to you about alternative medicine/complementary and integrative medicine			0.8773 ^2^			0.0001 ^4^
N	1488	80				
Mean (SD)	6.5 (3.35)	6.5 (3.64)		6.5 (2.55)	5.8 (9.58)	
Median (IQR)	7.0 (4.0, 10.0)	8.0 (4.0, 10.0)		7.0 (4.0–10)	6.0 (4.0–10)	

Body image/attractiveness/sexuality			0.3225 ^2^			0.0132 ^4^
N	1293	68				
Mean (SD)	6.4 (3.13)	6.0 (3.52)		6.4 (2.38)	5.9 (8.63)	
Median (IQR)	7.0 (5.0, 9.0)	6.0 (3.5, 9.0)		7.0 (4.0–9.0)	6.0 (4.0–9.0)	

Existential Issues (anxiety and/or questions regarding the meaning of life and/or your purpose in life)			0.5402 ^2^			0.6197 ^4^
N	1079	71				
Mean (SD)	6.3 (3.21)	6.1 (3.82)		6.3 (2.45)	6.4 (8.96)	
Median (IQR)	7.0 (4.0, 9.0)	7.0 (3.0, 10.0)		7.0 (4.0–9.0)	7.0 (4.0–10)	

Changes in your relationships with friends and/or family members			0.7054 ^2^			0.2676 ^4^
N	1057	63				
Mean (SD)	6.3 (3.28)	6.4 (3.90)		6.3 (2.50)	6.5 (9.33)	
Median (IQR)	7.0 (4.0, 9.0)	8.0 (3.0, 10.0)		7.0 (4.0–9.0)	8.0 (3.0–10)	

The impact of your cancer diagnosis on your employment status and your career			0.0565 ^2^			<0.0001 ^4^
N	904	23				
Mean (SD)	6.3 (3.53)	4.9 (4.63)		6.3 (2.69)	4.4 (9.22)	
Median (IQR)	7.0 (4.0, 10.0)	5.0 (0.0, 10.0)		7.0 (4.0–10)	2.0 (0.0–9.0)	

Cognitive/memory issues (i.e., brain fog or chemo brain)			0.3290 ^2^			0.3409 ^4^
N	1167	61				
Mean (SD)	6.3 (3.06)	5.9 (3.36)		6.3 (2.32)	6.1 (7.82)	
Median (IQR)	7.0 (4.0, 9.0)	6.0 (4.0, 9.0)		7.0 (4.0–9.0)	7.0 (4.0–8.0)	

Concerns as regards how loved ones will cope practically and emotionally if you pass away from breast cancer			0.5934 ^2^			0.0022 ^4^
N	1170	76				
Mean (SD)	6.3 (3.15)	6.1 (3.19)		6.2 (2.40)	6.8 (8.14)	
Median (IQR)	7.0 (4.0, 9.0)	6.0 (4.0, 9.0)		7.0 (4.0–9.0)	8.0 (5.0–10)	

Changes in intimacy with your partner			0.5182 ^2^			0.0314 ^4^
N	1164	51				
Mean (SD)	6.0 (3.20)	5.7 (3.83)		6.0 (2.43)	5.6 (9.16)	
Median (IQR)	6.0 (4.0, 9.0)	5.0 (2.0, 10.0)		6.0 (4.0–9.0)	5.0 (2.0–9.0)	

The impact of your breast cancer diagnosis on dating and/or socializing			0.2502 ^2^			0.2292 ^4^
N	756	29				
Mean (SD)	5.8 (3.58)	5.0 (4.61)		5.9 (2.73)	5.5 (11.3)	
Median (IQR)	6.0 (3.0, 9.0)	5.0 (0.0, 10.0)		6.0 (3.0–9.0)	5.0 (2.0–10)	

Your fertility			0.0019 ^2^			<0.0001 ^4^
N	367	11				
Mean (SD)	5.0 (4.26)	0.9 (3.02)		4.9 (3.23)	0.6 (4.84)	
Median (IQR)	5.0 (0.0, 10.0)	0.0 (0.0, 0.0)		5.0 (0.0–10)	0.0 (0.0–0.0)	

	Quality of life How much do you think the following research ideas have to improve quality of life for patients with breast cancer and their families (10-point scale, 0 = none, 10 = as much as I can imagine)					

Lifetime Access to Online Patient Educational Resources			0.2588 ^2^			0.3132 ^4^
N	1883	136				
Mean (SD)	8.4 (2.34)	8.2 (2.75)		8.4 (1.79)	8.5 (6.03)	
Median (IQR)	10.0 (8.0, 10.0)	10.0 (7.0, 10.0)		10 (8.0–10)	10 (8.0–10)	

A Breast Cancer Wellness Program for EBC and MBC patients			0.0926 ^2^			<0.0001 ^4^
N	1879	137				
Mean (SD)	8.4 (2.31)	8.0 (2.75)		8.4 (1.77)	7.9 (6.51)	
Median (IQR)	9.0 (8.0, 10.0)	10.0 (7.0, 10.0)		9.0 (8.0–10)	9.0 (6.0–10)	

A study focusing on educational, practical, emotional and holistic support for patients with MBC			0.5105 ^2^			0.0004 ^4^
N	1865	137				
Mean (SD)	8.4 (2.27)	8.2 (2.61)		8.4 (1.74)	7.9 (6.94)	
Median (IQR)	9.0 (8.0, 10.0)	10.0 (7.0, 10.0)		9.0 (8.0–10)	10 (6.0–10)	

A study in patients with MBC to make the transition from oncology treatment to palliative/hospice care as smooth as possible			0.7291 ^2^			0.0581 ^4^
N	1850	134				
Mean (SD)	8.3 (2.38)	8.3 (2.74)		8.3 (1.81)	8.5 (6.38)	
Median (IQR)	9.0 (8.0, 10.0)	10.0 (8.0, 10.0)		9.0 (8.0–10)	10 (8.0–10)	

A study to train oncology providers about the specific/different needs of EBC vs. MBC patients			0.3198 ^2^			0.0754 ^4^
N	1872	135				
Mean (SD)	8.1 (2.43)	8.3 (2.66)		8.1 (1.86)	8.3 (5.89)	
Median (IQR)	9.0 (7.0, 10.0)	10.0 (8.0, 10.0)		9.0 (7.0–10)	10 (8.0–10)	

Virtual Second Opinion Breast Cancer Clinic			0.9879 ^2^			0.5364 ^4^
N	1879	135				
Mean (SD)	8.1 (2.56)	8.1 (2.86)		8.1 (1.96)	8.0 (7.04)	
Median (IQR)	9.0 (7.0, 10.0)	10.0 (7.0, 10.0)		9.0 (7.0–10)	9.0 (8.0–10)	

A study focusing on educational, practical, emotional and holistic support for patients with EBC			0.8702 ^2^			0.0116 ^4^
N	1881	138				
Mean (SD)	8.1 (2.44)	8.1 (2.80)		8.1 (1.86)	7.8 (7.96)	
Median (IQR)	9.0 (7.0, 10.0)	10.0 (7.0, 10.0)		9.0 (7.0–10)	10 (6.0–10)	

A study focusing on educational, practical, emotional and holistic support for caregivers/family members of patients with MBC			0.8983 ^2^			0.0018 ^4^
N	1871	137				
Mean (SD)	7.9 (2.49)	7.9 (2.86)		7.9 (1.90)	7.5 (7.64)	
Median (IQR)	9.0 (7.0, 10.0)	10.0 (6.0, 10.0)		9.0 (7.0–10)	9.0 (5.0–10)	

A Complementary and Integrative Medicine Workshop promoting emotional wellbeing and addressing spiritual suffering			0.6165 ^2^			0.0025 ^4^
N	1873	136				
Mean (SD)	7.5 (2.78)	7.4 (2.96)		7.5 (2.11)	7.1 (6.99)	
Median (IQR)	8.0 (6.0, 10.0)	8.0 (5.0, 10.0)		8.0 (6.0–10)	8.0 (5.0–10)	

A study focusing on educational, practical, emotional and holistic support for caregivers/family members of patients with EBC			0.6659 ^2^			0.0002 ^4^
N	1873	137				
Mean (SD)	7.3 (2.73)	7.4 (3.10)		7.3 (2.08)	6.8 (8.52)	
Median (IQR)	8.0 (5.0, 10.0)	9.0 (5.0, 10.0)		8.0 (5.0–10)	8.0 (5.0–10)	

A couples workshop to address relationship stresses and intimacy issues in patients with EBC vs. MBC			0.4301 ^2^			<0.0001 ^4^
N	1869	133				
Mean (SD)	6.8 (3.06)	6.6 (3.60)		6.8 (2.33)	5.9 (8.85)	
Median (IQR)	8.0 (5.0, 10.0)	8.0 (5.0, 10.0)		8.0 (5.0–10)	7.0 (3.0–9.0)	

Reconnect oncology providers with a past patient			0.8533 ^2^			0.4414 ^4^
N	1869	133				
Mean (SD)	6.3 (3.21)	6.3 (3.55)		6.3 (2.44)	6.4 (8.59)	
Median (IQR)	7.0 (4.0, 9.0)	7.0 (4.0, 10.0)		7.0 (4.0–9.0)	6.0 (4.0–10)	

	Integrative medicine					

Have you ever seen an integrative medicine or complementary medicine provider for breast cancer symptoms or treatments, n (%)			0.0003 ^1^			<0.0001 ^3^
Yes	400 (20.9%)	11 (7.6%)		20.3% (19.4%–21.3%)	7.6% (7.2%–8.1%)	
No	1415 (73.9%)	122 (84.1%)		74.5% (73.3%–75.6%)	83.0% (82.1%–83.9%)	
Unsure	101 (5.3%)	12 (8.3%)		5.2% (4.9%–5.5%)	9.4% (8.8%–9.9%)	
Missing	319	57				

If so, did you ever take vitamins, minerals, herbal supplements or intravenous medications recommended by that provider, n (%)			0.0056 ^1^			0.1463 ^3^
Yes	174 (43.7%)	4 (36.4%)		43.1% (39.9%–46.4%)	47.3% (41.3%–53.3%)	
No	215 (54.0%)	5 (45.5%)		54.5% (51.2%–57.7%)	45.9% (39.9%–51.9%)	
Unsure	9 (2.3%)	2 (18.2%)		2.4% (2.1%–2.7%)	6.8% (5.3%–8.3%)	
Missing	1837	191				

If so, did your Oncology team know that you were taking these treatments, n (%)			0.0884 ^1^			<0.0001 ^3^
Yes	151 (87.3%)	2 (50.0%)		87.4% (85.1%–89.6%)	24.0% (17.7%–30.4%)	
No	9 (5.2%)	1 (25.0%)		4.8% (3.9%–5.8%)	70.4% (63.1%–77.7%)	
Unsure	13 (7.5%)	1 (25.0%)		7.8% (6.4%–9.3%)	5.5% (3.7%–7.4%)	
Missing	2062	198				

Saw integrative medicine or complementary medicine provider to strengthen immune system, n (%)			0.3369 ^1^			0.0885 ^3^
No	273 (68.3%)	6 (54.5%)		68.4% (65.5%–71.2%)	79.2% (75.2%–83.2%)	
Yes	127 (31.8%)	5 (45.5%)		31.6% (28.8%–34.5%)	20.8% (16.8%–24.8%)	
Missing	1835	191				

Saw integrative medicine or complementary medicine provider to support conventional medicine, n (%)			0.3688 ^1^			0.0642 ^3^
No	237 (59.3%)	8 (72.7%)		59.0% (55.8%–62.2%)	46.2% (40.2%–52.2%)	
Yes	163 (40.8%)	3 (27.3%)		41.0% (37.8%–44.2%)	53.8% (47.8%–59.8%)	
Missing	1835	191				

Saw integrative medicine or complementary medicine provider to combat side effects of cancer and traditional cancer treatments, n (%)			0.0267 ^1^			<0.0001 ^3^
No	158 (39.5%)	8 (72.7%)		40.1% (37.0%–43.3%)	74.6% (70.0%–79.1%)	
Yes	242 (60.5%)	3 (27.3%)		59.9% (56.7%–63.0%)	25.4% (20.9%–30.0%)	
Missing	1835	191				

Saw integrative medicine or complementary medicine provider because lack of confidence in traditional/ Western medicine, n (%)			0.4029 ^1^			0.6881 ^3^
No	384 (96.0%)	10 (90.9%)		95.6% (95.1%–96.2%)	96.7% (96.0%–97.5%)	
Yes	16 (4.0%)	1 (9.1%)		4.4% (3.8%–4.9%)	3.3% (2.5%–4.0%)	
Missing	1835	191				

Saw integrative medicine or complementary medicine provider because of other reason, n (%)			0.2631 ^1^			0.0072 ^3^
No	359 (89.8%)	11 (100.0%)		89.9% (88.8%–91.1%)	100% (100%–100%)	
Yes	41 (10.3%)	0 (0.0%)		10.1% (8.9%–11.2%)	0.0% (0.0%–0.0%)	
Missing	1835	191				

How satisfied were you with the care you received an integrative medicine or complementary medicine provider, n (%)			0.0002 ^1^			<0.0001 ^3^
Very satisfied	201 (50.9%)	4 (40.0%)		50.6% (47.3%–53.9%)	31.2% (26.0%–36.5%)	
Satisfied	135 (34.2%)	4 (40.0%)		34.2% (31.3%–37.2%)	61.4% (55.7%–67.2%)	
Neither satisfied or dissatisfied	48 (12.2%)	0 (0.0%)		12.5% (11.0%–13.9%)	0.0% (0.0%–0.0%)	
Dissatisfied	10 (2.5%)	1 (10.0%)		2.5% (2.2%–2.8%)	3.7% (2.8%–4.5%)	
Very dissatisfied	1 (0.3%)	1 (10.0%)		0.3% (0.2%–0.3%)	3.7% (2.8%–4.5%)	
Missing	1840	192				

	Medical Second Opinion					

Have you ever received a second opinion regarding your breast cancer diagnosis and treatment plan, n (%)			<0.0001 ^1^			<0.0001 ^3^
Yes	792 (41.5%)	29 (20.0%)		40.9% (39.5%–42.3%)	21.7% (20.6%–22.9%)	
No	1102 (57.8%)	114 (78.6%)		58.4% (57.0%–59.9%)	77.5% (76.4%–78.7%)	
Unsure	14 (0.7%)	2 (1.4%)		0.7 % (0.7 %–0.7 %)	0.7 % (0.7 %–0.8 %)	
Missing	327	57				

If so, did you find it beneficial, n (%)			0.0596 ^1^			<0.0001 ^3^
Yes	752 (95.9%)	25 (92.6%)		95.9% (95.6%–96.3%)	83.7% (81.6%–85.8%)	
No	21 (2.7%)	0 (0.0%)		2.7 % (2.4 %–2.9 %)	0.0 % (0.0 %–0.0 %)	
Unsure	10 (1.3%)	2 (7.4%)		1.3 % (1.2 %–1.4 %)	16.3% (14.2%–18.4%)	
Not applicable	1 (0.1%)	0 (0.0%)		0.1 % (0.1 %–0.1 %)	0.0 % (0.0 %–0.0 %)	
Missing	1451	175				

Have patient advocates ever assisted you with decision making and the logistics of breast cancer treatment, n (%)			0.0029 ^1^			<0.0001 ^3^
Yes	371 (19.4%)	28 (19.4%)		19.3% (18.4%–20.2%)	25.7% (24.4%–26.9%)	
No	1371 (71.7%)	91 (63.2%)		71.8% (70.6%–73.0%)	57.2% (55.6%–58.8%)	
Unsure	169 (8.8%)	25 (17.4%)		8.9 % (8.4 %–9.4 %)	17.1% (16.2%–18.1%)	
Missing	324	58				

Clinical Trials						
Since being diagnosed with breast cancer, have you participated in a clinical trial, n (%)			<0.0001 ^1^			<0.0001 ^3^
Yes	907 (47.2%)	60 (40.5%)		46.2% (44.7%–47.7%)	47.3% (45.6%–48.9%)	
No	822 (42.8%)	54 (36.5%)		43.2% (41.7%–44.6%)	29.4% (28.1%–30.8%)	
Unsure	192 (10.0%)	34 (23.0%)		10.6% (10.1%–11.2%)	23.3% (22.1%–24.5%)	
Missing	314	54				

If you have not participated in a clinical trial for breast cancer patients, would you be open to doing so in the future, n (%)			0.0001 ^1^			<0.0001 ^3^
Yes	612 (62.0%)	35 (40.7%)		61.6% (59.7%–63.6%)	45.5% (43.3%–47.8%)	
No	84 (8.5%)	16 (18.6%)		8.5 % (7.9 %–9.2 %)	10.2% (9.4 %–11.1%)	
Unsure	291 (29.5%)	35 (40.7%)		29.8% (28.1%–31.5%)	44.2% (42.0%–46.5%)	
Missing	1248	116				

^1^ Chi-Square *p*-value; ^2^ Two sample *t*-test; ^3^ Propensity score-weighted chi-square *p*-value; ^4^ Propensity score-weighted *t*-tests.

**Table 3 cancers-16-02494-t003:** Qualitative Data from Survey Respondents Aged 80+ Years.

Category	Feedback/Comment(s)	Patients’ Age at Survey Completion
Breast Cancer Education	“Don’t overwhelm patient with printed materials. Much is duplication and much does not relate to patient and patient’s condition”	81
	“I think this depends on the individual patients. Some do not want to have so much information that they just feel confused about making a decision. They just want a more directed approach. Then there are others who have the opposite views and need to get all the info that is available”	83
	“I would have liked to have more information regarding the impact of genetic testing on such issues as insurance factors for my daughters. I would have liked to participate in genetic testing if insurance issues had not been a factor”	83
	“Develop a breast cancer app that includes info on the individual that would be immediately available to be shown to a new provider. Connect it to online resources and new information the patient can access over time”	80
	“Better non-medical terms information about the pros and cons of chemotherapy before or after surgery, and the choices between breast removal and lumpectomies. From what I hear from current patients, the providers are telling patients what they should do instead of giving them complete explanations of choices and the pros and cons”	84
Side-Effects of Breast Cancer Treatment	“Would have liked to have known the effects of radiation on breast prior to having radiation…. Would have liked to have had discussion of lymphedema prior to surgery again just to prepare me before I got it in my breast not arm”	80
	“I would suggest much more detailed information concerning side affects of radiation. The information provided does not share the possibility of the burning of the skin-simply states slight sunburn…”	81
	“inform them more about possibility of lymphademia (sic). I have it.	84
	“…I should have been told the difficulty of reconstruction and also been told that because of previous radiation treatments for breast cancer that the skin and tissues were damaged and that there was a chance the implant would not hold. I went through the treatments only to have the implant fall and needed to be removed”	80
	“I am not sure if the pills I took for the five years following surgery were estrogen inhibitors or not, but they did affect my skeleton. That is the only aspect of treatment that truly upset me, since I had a very strong skeleton for my age and developed osteopenia because of the medicine….”	80
Survivorship/Long Term Sequelae of Breast Cancer Treatment	“Focus on post surgical mastectomy and what is available as a prosthesis and where items can be purchased”	86
	“I don’t like to think of my self as a "breast cancer patient.” I had breast cancer, and I may have it again, but it doesn’t define me”	80
	“The first year check up was far more painful than I thought it would be”	83
Emotional Support During and After Diagnosis	“Connect breast cancer survivors with groups/activities in the community”	80
	“Patients need an ongoing source of physical and emotional assistance which they can rely on from the beginning to the end of their journey of healing”	82
	“Access to network of support persons. I live in [redacted] where people in health care had no suggestions. I was new to the community and did not know what to do. I needed to find someone who experienced my situation”	83
	“Discussing it with family. My family does not want to discuss it at all. It’s like if they don’t talk about it, it does not exist”	80
	“Continue open discussions and followups like this”	86
	“Training for the nurse practitioner to enable her/him to emotionally support and care for the patient, even as the years post op extend toward five”	80
	“Use of volunteers and survivors for group or individuals”	81
Care Concerns and Provider Sensitivity	“View each patient individually and try to treat at the level they are comfortable”	83
	“What I am most " upset" about is the doctor who informed me I had cancer!!”	81
	“My experience with Radiology ….I complained of the positioning which was causing my neck to hurt. I was told there was nothing they could do. Eventually, they did listen and apologized and adjustments were made…”	81
	“The oncologist …was not interested in the pain I was receiving from radiation.	88
	“Cooperation among hospitals/providers is important”	86
	“I was concerned that they seemed to be delaying getting treatment started and I was anxious about that delay….I was confused by all the diagnostic testing”	84
	“I think the emotional impact is probably related to age. If I were younger, I think I would have a much different perspective. That said, it bothers me a little to hear references to my age and longevity aspects”	80
Diversity, Ethnicity and Inclusion	“What is the percentage of Hispanic women patients over the age of 79 that gets care at the Breast Clinic?”	81
	“As a Hispanic woman…. I have seen a difference in the care I have received. It is a subtle change either due to my age or other factors such as less staff or staff that is less attentive for whatever reason”	81
Access to Breast Cancer Care	“Early surgical treatment should be available after diagnosis is made”	83
	“Faster mammogram reports”	82
Screening for Patients With Dense Breast Tissue	“I have dense breast tissue so I think women with this should have both the mammogram and MBI. For first years after surgery and treatment had both mammogram and MBI. Now these last few years alternate. I have mammogram one year and then the MBI the next.”	86
	“Better insurance coverage for patients with dense breasts and further options for evaluation and treatment”	84
Diet and Exercise	“Have diet and exercise be a very important part of care”	82
	“I believe help from a dietitian to strongly suggest food planning that would be more beneficial to the recovery of the patient”	82
Sexual Dysfunction	“Help with sexual dysfunction”	82
Research Advocacy	“Focus on a cure and preventing Breast Cancer”	81
	“I believe in continued research”	82
	“Just participating in a clinical trial and getting the results is very satisfying”	81
	“How do genetics affect breast cancer in the families of cancer pts?”	82
	“Hopefully, research will develop a vaccine to help prevent breast cancer. And research will help discover the causes of breast cancer”	81

## Data Availability

The datasets for this study are available from the corresponding author upon request.

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
