# Peer review of "Advocate-BREAST80+: A Comprehensive Patient and Advocate-Led Study to Enhance Breast Cancer Care Delivery and Patient-Centered Research in Women Aged ≥80 Years"

_cancers, 2024, doi:10.3390/cancers16142494_

Round 1

Reviewer 1 Report (New Reviewer)

Comments and Suggestions for Authors

The study shows results from the Advocate-BREAST80+, which examines the experiences of breast cancer patients aged 80 and older (P80+) compared to those under 80 (P80-).

Conducted at the Mayo Clinic, this survey study revealed significant differences in treatment recommendations, symptom severity, and satisfaction with care.

P80+ patients were less likely to be advised to undergo surgery, radiation, and systemic therapies but were equally likely to follow through when recommended. They reported lower satisfaction with information on treatment side effects and expressed distinct educational needs.

Congratulations to the authors of the manuscript. I think the study is original and evaluates an unexplored topic. I just have one comment: as the authors state, very often P80+ patients are under- or over-treated. This is also the case of breast cancer recurrences. Many times, old patients are offered mastectomy even in case when breast conservation is possible. Please cite this study PMID: 33431329 to improve the quality of your manuscript.

Author Response

Please see attachment, thank you.

Reviewer 2 Report (New Reviewer)

Comments and Suggestions for Authors

In this manuscript, the authorsdid an online survey and collected data including a 23 page questionnaire (containing 147 questions) to assess patients' concern sand satisfaction on breast cancer diagnosis and treatment. The results are important and the manuscript is well organized. I recommend its publication on Cancers after the following minor issues are considered:

1.It would be advisable to further clarify the specific aims and significance of this research to facilitate reader's comprehension of its importance at the beginning of the document.

2. More transitional statements between paragraphs were supposed to be added to enhance the logical flow and coherence of the manuscript.

3. The quantity of samples is limited. Please augment the number of samples significantly to minimize the error and enhance the reliability and accuracy of data.

4. I wonder whether there are other factors that affect patient's thoughts, for instance, family environment.

5. To delve into data analysis, exploring the disparities among different age groups and the potential reasons behind them are necessary.

Comments on the Quality of English Language

Minor editing of English language required

Author Response

Reviewer 3 Report (New Reviewer)

Comments and Suggestions for Authors

The study aims to compare the needs of breast cancer patients under and over 80 years old in regard with their perception of the management of their disease.

The study is thoroughly conducted and it is of interest for a focused management of breast cancer patients over 80 years.

The manuscript is clear and relevant for the field of interest and is presented in a structured manner. It presents information useful for specific interventions to enhance continuity of care, communication, holistic care, and long-term psychosocial support for this category of patients.

The method and the results are clearly presented. The discussions are focused on the main aim of the study and the conclusions are well formulated.

The strengths and the limitations of the study are highlighted adequately, proving professionalism in conducting the study.

The references are recent and relevant.

The study might attract a wide readership, taking into consideration the interest of professionals in a better and focused management of breast cancer patients, based on their needs.

Author Response

This manuscript is a resubmission of an earlier submission. The following is a list of the peer review reports and author responses from that submission.

Round 1

Reviewer 1 Report

Comments and Suggestions for Authors

Dear researchers, please organize the tables and results better, understanding them becomes complicated, describe the biases and how you have solved them. Indicate the ethics committee where the study was presented and the data protection regulations followed in your study. As well as the manuals of good research practices that they have followed.

Author Response

Reviewer 1

Comments and Suggestions for Authors

Comment: Please organize the tables and results better, understanding them becomes complicated.

Response: Thank you for this suggestion. We have detailed the edits that we made to present all of our results as succinctly as possible in the Tables and manuscript in our responses to reviewer 2’s comments below.

Comment: Describe the biases and how you have solved them.

Response: Many thanks for your thoughtful feedback.  We have described potential sources of bias in this project in the second paragraph of our discussion.

Comment: Indicate the ethics committee where the study was presented and the data protection regulations followed in your study. As well as the manuals of good research practices that they have followed.

Response: Information as regards Ethics Committee Review, data protection rules, and good research practices followed are already stated in the body of the manuscript, at the end of the first paragraph in page 4, as follows: “The Mayo Clinic Institutional Review Board (IRB1815-04) reviewed and approved this study. Data were handled in a manner consistent with both US laws and the Declaration of Helsinki”. This information is also included in the Statements and Declarations Section of the Manuscript, as follows:

Ethics Approval: This research was conducted in accordance with the Declaration of Helsinki, and the study protocol was reviewed by the Mayo Clinic Institutional Review Board (IRB 1815-04).

Consent to Participate: Informed consent was obtained from all participants prior to enrollment to the registry.”

Reviewer 2 Report

Comments and Suggestions for Authors

This is a very important and needed topic in a population seeing an increase in individuals ages 80+ years. That being said, the statistical approach needs to be improved.

1. Since the weighting system is in place, the descriptive statistics should be presented using weighted percent and 95% CI for the weighted percent, the weighted median and weighted interquartile range (definitely not mean and standard deviation, unless the authors tested and are confident regarding the normal distribution of the continuous variables). Instead of chi-square tests the authors should run the Rao-chi-square tests.

2. Since age is a non-modifiable factor, it should not be used as the outcome under investigation. Age group can be tested for effect modification of a certain exposure-outcome association. If found to be significant, subgroup analyses should be performed by age-group levels (dichotomous in this case), while still including age as a continuous variable in the model. Therefore the analyses performed at this point are incorrect/not valid.

3. If age/age-group is used as the outcome under investigation, no confounders can be used in the models, since no variables would satisfy the definition of a confounder, because there is no variable that can be a risk factor for age. Once again, age cannot be modified.

4. The selection of confounders for a specific exposure-outcome association should be done based on prior literature. The minimum sufficient sent of variables to be included in the analyses should be selected via a directed acyclic graph (DAG) methodology. Propensity score matching technique should not be used as the sole method for addressing confounding. 

Reviewer 3 Report

Comments and Suggestions for Authors

The questionnaire results are too detailed and you should make it shorter by dropping the less important questions or joining several questions in groups.

Author Response

Reviewer 3

Comment: The questionnaire results are too detailed and you should make it shorter by dropping the less important questions or joining several questions in groups.

Response: We have attempted to present the results as clearly as possible in the manuscript and Tables, incorporating feedback from all of our reviewers. Although we agree that the questionnaire was somewhat long, we are concerned that dropping or consolidating responses may not fully honor the time and effort that our ~2,500 participants spent providing these data. Therefore, we would prefer to be able to publish all of the questionnaire data. However, if you strongly prefer that we consolidate/abbreviate these questionnaire results, we would appreciate more specific guidance regarding i) results that should be retained vs. omitted, as well as ii) how we should reclassify/join questions.

Reviewer 4 Report

Comments and Suggestions for Authors

This is a very good research work on the cancer care received by older women. As the number of breast cancer diagnoses in older age groups increases, the type and the level of care of this population groups must meet hteir specific needs and expectations and this work is in the right direction.

My main comment though is that it should be emphasised that this study samle is highlty selective and does not by any  means represent the average breast cancer patient. There is some mention of the generalizability of the results in lines 310-311, but this should be mentined more emphatically.

A paragraph on limitations of the study should be added. Issues like vulnerability, insurance coverage and educational level should be considered before reaching conclusions. The approach needs to be tailored to the secific needs of the target population and this needs to come clearly across to the reader in this paper.

Author Response

Comment: This is a very good research work on the cancer care received by older women. As the number of breast cancer diagnoses in older age groups increases, the type and the level of care of this population groups must meet their specific needs and expectations and this work is in the right direction.

Response: Many thanks, we appreciate your review of our work.

Comment: My main comment though is that it should be emphasized that this study sample is highly selective and does not by any means represent the average breast cancer patient. There is some mention of the generalizability of the results in lines 310-311, but this should be mentioned more emphatically.

Response: Thank you. We have modified the text in this section as follows: “Further, as most survey respondents were married/widowed and Caucasian Christians living in the Midwest of the US, it is important to note that our conclusions may not be generalizable. Therefore, a future study should enroll a more ethnically, racially, and geographically diverse cohort of BC P80+.”

Comment:  A paragraph on limitations of the study should be added. Issues like vulnerability, insurance coverage and educational level should be considered before reaching conclusions.

Response: Thank you for your comment. Limitations of the study are addressed in the second paragraph of the discussion. We have considered these variables in our analyses.

Comment: The approach needs to be tailored to the specific needs of the target population and this needs to come clearly across to the reader in this paper.

Response: Thank you for your comment. We have updated the second paragraph on page 9 to highlight the need for individualized care for this patient population.

Round 2

Reviewer 2 Report

Comments and Suggestions for Authors

In the updated version of the paper:

1. Table 2 presents weighted mean and standard deviation instead of weighted median and IQR. This needs to be corrected.

2. Although the differences between Chi-square and Rao chi-square may be minute for this particular paper, the correct approach for weighted analyses is to present the Rao chi-square instead of the regular chi-square values, and therefore in this paper the Rao chi-square values should be reported instead.

3. There are no factors modifying age/risk factors for age. Age can be evaluated in a simple correlation with the outcome under investigation, but not as an exposure or even as an outcome in a multivariable analysis. You can use age as a starting point in an SEM, but not as an exposure in a multivariable analysis. You can test the other variables for potential effect modification of the association between age and the outcome under investigation, but not to include them in a multivariable model as independent variables. In fact, the diagram drawn in the response document is not valid, since race/ethnicity is not a risk factor for age, social factors are not risk factors for age, etc.

4. In the examples presented by the authors, they stated that germline genetic haplotype is a non-modifiable factor. This statement is incorrect. There are emerging technologies in the field of genetics, such as CRISPR-Cas9 gene editing, that can modify germline genetic material.